# The Roles of Vitamins in Diabetic Retinopathy: A Narrative Review

**DOI:** 10.3390/jcm11216490

**Published:** 2022-11-01

**Authors:** Varis Ruamviboonsuk, Andrzej Grzybowski

**Affiliations:** 1Department of Biochemistry, Faculty of Medicine, Chulalongkorn University, Bangkok 10330, Thailand; 2Department of Ophthalmology, University of Warmia and Mazury, 10-719 Olsztyn, Poland; 3Institute for Research in Ophthalmology, Foundation for Ophthalmology Development, 61-166 Poznan, Poland

**Keywords:** vitamins, nutrients, dietary supplements, diabetic retinopathy

## Abstract

There have been attempts to evaluate the roles of vitamins for the prevention or treatment of eye conditions, such as glaucoma, age-related macular degeneration, and diabetic retinopathy (DR). Regarding DR, two main groups of studies can be identified. The first group focuses on the association between serum levels of an individual vitamin and DR. Many studies have found that lower serum levels of vitamins, particularly vitamin D, are significantly associated with the development, or severity, of DR, while some studies have not supported this trend. The second group evaluates dietary vitamin intakes and DR. A small, randomized placebo-controlled trial did not show any benefit of vitamin E intake on improving the area of retinal hemorrhage or diabetic macular edema at 12 months. A pilot study of patients with mild-to-moderate non-proliferative DR received tablets of combined vitamins B6, B9, and B12 for 6 months and significant improvement in retinal sensitivity and retinal thickness resulted. Two large prospective cohorts showed that high dietary intake of vitamin B6, and fruit rich in vitamin C and E, could significantly lower the risk of DR by 50% after an eight-year follow-up. Properly designed, randomized controlled trials are needed to support the results.

## 1. Introduction

Diabetes mellitus (DM), a chronic non-communicable disease, characterized by chronic hyperglycemia [1], is one of the leading global health challenges. Patients with diabetes have increased risk of developing both macrovascular and microvascular complications, which can lead to morbidity and mortality of patients with DM [2]. In 2021, an estimated 6.7 million mortalities worldwide were caused by diabetes. Visual impairment and blindness are morbidities caused by diabetic retinopathy (DR), an ocular complication of diabetes. In 2020, DR was the cause of moderate or severe visual impairment in approximately 3.2 million people worldwide, among them, 0.9 million adults aged 50 years and older were considered blind [3]. The prevalence of DR was estimated at an average of a third of patients with DM. Since more than 783 million people are predicted to have DM in 2045 [4], more than 250 million people are expected to have DR.

DR is a microvascular complication of DM. caused by dysfunction of endothelial cells of retinal capillaries due to long-term exposure to hyperglycemia [5]. There are several known risk factors of developing DR, such as hyperglycemia, hypertension, dyslipidemia, diabetes duration, Hispanic or South Asian race, pregnancy, puberty, and cataract surgery. The pathophysiology of DR is believed to be associated with chronic exposure to hyperglycemia which can initiate biochemical and physiological changes, and can lead to microvascular damage and retinal dysfunction. Biochemical changes include accumulation of sorbitol and advanced glycation end-products, oxidative stress, protein kinase C activation, inflammation, and upregulation of the renin–angiotensin system and vascular endothelial growth factor (VEGF) [5], induced by hypoxia and advanced glycation end-products, and can ultimately lead to neovascularization in proliferative diabetic retinopathy (PDR) [6]. On the physiological side, retinal arteriolar dilatation might be an indicator of microvascular dysfunction, which could lead to capillary wall dilatation, leakage, and rupture [5]. These biochemical and physiological changes could be the target for treatment of DR.

Vitamins are essential nutrients and antioxidants which play a large part in metabolisms in human bodies. They are used as dietary supplements to improve or maintain overall health [7]. Although they may not be classified as drugs, some vitamins are recommended for prevention or treatment of diseases. The common use of vitamins as a treatment modality pertains to conditions in which the daily requirement of an essential nutrient is deficient. Vitamin D deficiency, for example, is one of the most prevalent medical conditions, affecting 14–59% of adult populations worldwide. This condition results in poor bone development and general ill health, and is prevented and treated with vitamin D. The use of vitamin B9 for prevention of severe birth defects [8], and vitamin B1 for prevention of Wernicke encephalopathy [9], and other uses of vitamins in the prevention, or treatment, of medical conditions, is well documented.

There have been attempts to evaluate the roles of vitamins in prevention and treatment of conditions other than vitamin deficiency. For eye diseases, the focus is on conditions associated with oxidative stress, such as age-related macular degeneration (AMD), and glaucoma, including DR. For AMD, a large scale, multicentered, randomized, controlled clinical trial, commonly known as an Age-Related Eye Disease Study (AREDS), evaluated whether supplements of multivitamins could reduce the risk of development of late stage AMD in [10]. A recent post-hoc analysis from AREDS concluded that higher dietary intake of multiple nutrients, minerals, carotenoids, and vitamins were associated with a decreased risk of late AMD [11].

There are rationales for using vitamins for prevention or treatment of DR, since some vitamins have been shown to have positive effects on increasing the elasticity of blood vessels and endothelial function [12,13,14], whereas some may have an inhibitory effect on the retinal neovascularization found in late stages of DR. The aim of this review was to assess the current situation of studies on vitamins in regard to DR and to explore the results, so as to lay the ground for future research on vitamins for DR.

## 2. Methodology

We searched PubMed and Google Scholar databases for articles published between January, 2011, and April, 2022, using the following query terms: (vitamins) AND (diabetic retinopathy). We found 506 articles. After abstract screening, all articles that were not available in English, all studies on animals, in vitro models, review articles, systematic reviews and meta-analyses were excluded. In total, 49 studies focusing on the roles of vitamins on DR were retrieved for full text and thorough review, as shown in Figure 1.

## 3. Results

The majority of studies on the use of vitamins in regard to DR in the literature were aimed at finding the association between serum levels of individual vitamins and the development or progression of DR. All these studies were cross-sectional, designed to determine serum levels of vitamins in patients with different DR severity levels, including those without DR. There were fewer studies on vitamins for DR which aimed at assessing dietary vitamin supplements, in terms of the amount of daily intake of vitamins, and the development or progression of DR. There were two small studies aimed at the use of vitamins as drugs for the treatment of DR. There were two large studies, following more than 900 participants for 8 years, which aimed at finding the association between dietary intake of vitamins and DR development. Studies on circulating vitamin levels, and dietary intake of vitamins and DR are summarized in Table 1 and Table 2, respectively.

### 3.1. Levels of Circulating Vitamins and Diabetic Retinopathy

Vitamin A, or retinol, has been shown to play roles in macromolecule metabolism, such as carbohydrate metabolism [15]. Isomers of vitamin A, all-trans-retinol and 11-cis-retinol, are important nutrients for retinal photoreceptors, which are essential for visual function [16]. A study by Rostamkhani et al. showed that the mean serum vitamin A level in patients with proliferative diabetic retinopathy (PDR) was significantly lower than for patients with diabetes without any eye diseases; this higher serum level reduced the risk of the development of DR by 31.1% [17]. Zhang et al., however, found that the mean serum vitamin A level in patients with DR was significantly higher than those with diabetes without DR. Moreover, they did not find significant difference in the serum levels between those with DR and those in the healthy control group [18].

Vitamin B1 (thiamine) is converted to an active derivative, thiamine pyrophosphate (TPP), which is a cofactor of transketolase (Tk), pyruvate dehydrogenase and α-ketoglutarate dehydrogenase enzymes. These enzymes are involved in glucose metabolism [19]. Vitamin B2 (riboflavin) is a cofactor for synthesizing L-methylfolate, which would affect the homocysteine level [20]. Vitamin B3 (niacin), at high dose, may result in increased risk of insulin resistance [21]. Vitamin B6 (pyridoxine) acts as a coenzyme in the pathway forming cystathione from homocysteine [22]. Vitamin B9 and B12 are essential coenzymes in the regulation of the homocysteine pathway. An elevated level of homocysteine, which could be caused by depletion of vitamin B9 (folic acid) and B12 (cobalamin), is a risk factor in the development of vascular diseases, and its association with an increased risk of DR has been shown [23,24].

A study by Satyanarayana et al. found that, in comparisons between patients with DM and healthy controls, the mean serum levels of both vitamin B1 and B2 were not significantly different, whereas the mean serum levels of vitamins B6, B9 and B12 in patients with DM were significantly lower. In the same study, in comparisons between patients with DR and those with DM without DR, only serum vitamin B12 level was significantly lower in patients with DR, whereas there were no significant differences in mean serum vitamin B1, B2, B6 and B9 levels [25]. Cinici et al. measured serum levels of Thiamine pyrophosphate (TPP), an active derivative of vitamin B1, in patients with mild to moderate NPDR, severe NPDR, or PDR and found significantly lower levels of TPP in these groups of patients, compared to a group of healthy controls. However, the TPP levels in patients with any DR severity were not significantly lower than patients with DM without DR [26].

In another study, mean serum levels of vitamin B9 in patients with NPDR or PDR were found to be significantly lower than for patients with DM without DR and healthy controls; it was further found that the levels in the patients with PDR were also lower than those with NPDR [27]. A couple of studies showed that the mean serum level of vitamin B12 in patients with DR was significantly lower than those with DM without DR [25,28]. On the other hand, another study showed mean serum levels of vitamin B9 and B12 were not significantly different among patients with no DR, NPDR, PDR and healthy controls [29].

Vitamin C (ascorbic acid) is an antioxidant that provides beneficial effects on endothelial integrity which would result in improving endothelial dysfunction [30]. It has been shown to have a potential effect on reducing the progression of AMD [10,31]. In a study by Longo-Mbenza et al., the mean serum level of vitamin C in patients with DM was shown to be significantly lower than those without DM; in addition, the level in those with DR was significantly lower than those with DM without DR [32]. Two studies also found that the mean serum level of vitamin C in patients with DR was significantly lower than the level in patients with DM without DR, and healthy controls [33,34]. However, a study by Lam et al. found no significant difference of mean serum levels of vitamin C among patients with DM with NPDR, PDR and no DR [35].

Vitamin D (calcitriol) has been demonstrated to have roles in animal and in vitro studies, such as lowering intracellular ROS and VEGF expression in retinal cells [36], decreasing cell damage from high glucose levels in animals [37], and being a potent inhibitor of retinal neovascularization in animal subjects [38].There are several forms of vitamin D, but the primary circulating form of vitamin D is serum 25-hydroxy-vitamin D3 (25(OH)D), and this is widely accepted as the indicator of vitamin D status [39]. Vitamin D insufficiency is defined as having a serum level of 25(OH)D below 30 ng/mL, whereas vitamin D deficiency (VDD) is defined as having a serum level of 25(OH)D below 20 ng/mL [40]. Some studies have shown that the prevalence of VDD is higher in patients with type 2 diabetes than in healthy controls [41,42,43]. The results from many studies showed that patients with DR had significantly lower serum vitamin D levels compared with patients with DM without DR [32,44,45,46,47,48,49]. In addition, VDD was found to be associated with a higher prevalence of complications in patients with diabetes, such as neuropathy, nephropathy, and retinopathy [43,50]. Some studies found patients with DM and VDD had an increased risk of developing DR [47,51,52,53,54]. Another study also found higher odds of developing DR in patients with type 1 diabetes and VDD [55].

Ahmed et al. measured different forms of vitamin D in the serum and found that there was significant association between lower serum levels of either 1.25(OH)_2_D_3_ or 25(OH)D_3_, and DR (*p* = 0.006 and *p* = 0.03, respectively), whereas there was no significant association between the level of 24,25(OH)_2_D or 24,25(OH)_2_D_3_ and DR [56]. Ashinne et al., on the other hand, aimed to determine if different DR severity levels were associated with different serum vitamin D levels. They found significant difference in the mean serum levels of vitamin D among different DR severities. Patients with no DR, non-sight threatening DR, and sight threatening DR had mean serum levels of vitamin D at 13.7 ± 2.1 ng/mL, 12.8 ± 2.1 ng/mL, and 11.1 ± 2.2 ng/mL, respectively (*p* < 0.001) [45]. On the contrary, many other studies did not show significant difference in mean serum vitamin D level between patients with DR and DM without DR [28,42,57,58,59,60], and among patients with different DR severities [46,48,61,62]. Another study of patients with type 1 DM showed no significant difference between the mean serum level of 25(OH)D in those with DR and without DR [63].

Vitamin E, tocopherol and tocotrienol, is considered to be one of the most potent antioxidants and has been shown to suppress angiogenesis, and decrease oxidative stress [64,65]. Some studies showed that mean serum vitamin E level in patients with DR was significantly lower than those without DR [32,34,66], but a study by Lam et al. showed no significant difference in the serum vitamin E levels between patients with no DR and any DR severities [35].

In an overview of these studies of serum levels of individual vitamins and DR, there was a trend for there to be significantly higher levels of individual vitamins in healthy controls than in patients with DM, in patients with DM without DR than in patients with DR, and in patients with lower severity levels of DR than in those with higher severity levels of DR. However, the results in many studies did not support this trend.

### 3.2. Dietary Vitamins and Diabetic Retinopathy

There was a randomized controlled trial (RCT) that compared the effect of dietary vitamin E with placebo on the progression of DR. In this RCT, Ho et al. randomized 55 patients with NPDR to receive either tocotrienol-rich vitamin E supplement of 200 mg twice a day or placebo for 12 months. The primary outcome measure was the percentage of change in retinal hemorrhage and diabetic macular edema (DME), while the secondary outcome was the change in serum VEGF level. After 12 months, the area of retinal hemorrhages significantly increased, by 23.42%, in the placebo group, while the area in the vitamin group had no significant change. The area of DME, on the other hand, was significantly decreased, by 48.38% in the vitamin group, while there was no significant change in the placebo group. For serum VEGF level at 12 months, there was no significant difference between the two groups, or when compared with baseline in each group [67].

Regarding the dietary vitamin combination tablets commonly found in markets, a pilot study was conducted by Smolek et al. on a tablet combining vitamin B6, B9, and B12 and its association with DR. This study studied 10 patients with mild to moderate NPDR in both eyes. The patients were instructed to take the combination tablet twice daily for 6 months, so as to measure the percentage change in mean retinal sensitivity threshold and mean central retinal thickness. The results showed that this combination tablet increased mean retinal sensitivity by approximately 7% in the initial two months, followed by a plateau effect for the next three months. For the mean central retinal thickness, there was a decline in central retinal thickness by approximately 1–2% after 6 months, compared with baseline, and this trend did not show a plateau, as was found in the analysis of retinal sensitivity [68].

Most studies on individual dietary vitamins used a food frequency questionnaire (FFQ) to determine the level of vitamin intake per day or per week for analysis. An FFQ consists of a list of foods and allows participants to determine how often each item of foods is consumed over a period of time. Then, the frequency estimates of food intake are converted to an appropriate amount of nutrients consumed by using a nutrients database or food composition database (FCD), which provides the nutritional composition of foods [69,70]. The number of nutrients included in the FCD might differ from one to another depending on the requirement by each country or organization involved [70].

Compared with studies on serum levels of individual vitamins and DR, studies on dietary intake of individual vitamins and DR were fewer in number of publications. These studies are summarized in Table 2. For dietary vitamin A intake, Zhang et al. used an FFQ, calculated using the Nutrition System of Traditional Chinese Medicine Combining with Western Medicine, to determine intake per day. The authors found that per 100 µg/day of dietary vitamin A intake there was a lower risk for the development of DR in patients with DM of 17% (OR = 0.83, 95%CI = 0.70–0.98, *p* = 0.032) [18].

For dietary vitamin B supplements, a study by Horikawa et al. assessed the intake of vitamin B6 and the incidence of DR as part of the Japan Diabetes Complications Study (JDCS), which was a multicenter, nationwide cohort study of patients with type 2 diabetes across universities and general hospitals in Japan. A total of 978 patients without baseline retinopathy responded to the FFQ and were included in this study. Mean vitamin B6 intake was 1.4 mg/day, and quartiles were 1.09, 1.33, and 1.62 mg/day. The patients were divided into four groups according to quartiles. After 8 years, 79.0% of patients were still in the follow-up period. It appeared that the hazard ratios (HRs) for the 8-year incidence of DR in the second, third, and fourth quartiles of vitamin B6 intake, compared with the first quartile, were 1.17 (*p* = 0.403), 0.88 (*p* = 0.550), and 0.50 (*p* = 0.010), respectively. In summary, those who had an average consumption of 2.0 mg/day of vitamin B6 might be associated with an approximately 50% lower risk of developing DR than those who consumed 0.9 mg/day for 8 years [71].

In another study, which was also a part of the JDCS, conducted with the same cohort of 978 patients without retinopathy at baseline with 8 years of follow-up, fruit intake was assessed for DR development. Mean fruit intake at baseline of the patients in the quartiles ranged from 23 to 253 g/day. Multivariate-adjusted HRs for the second, third, and fourth quartiles of fruit intake compared with the first quartile for the 8-year incidence of DR were 0.66, 0.59, and 0.48 (test for trend, *p* < 0.01). This meant that the consumption of an average of 253 g of fruit per day might result in a 50% lower risk of DR compared with those consuming an average of 23 g/day. The authors suggested the potential involvement of vitamin C, carotene, retinol equivalent, and dietary fiber in DR development [72]. 

Another study by Sasaki et al. was aimed at assessing the associations of dietary intake of polyunsaturated fatty acids (PUFA) with DR. Increased PUFA intake was found to be associated with a reduced likelihood of the presence and severity of DR in patients with well-controlled diabetes. However, the authors also found no association between vitamin C, vitamin E, or b-carotene intake levels and the presence of DR in the whole sample [73].

A study by Alcubierre et al. assessed both serum levels and dietary intake of vitamin D and found that while low serum levels, or vitamin D deficiency, was associated with the presence and severity of DR, the dietary intake of vitamin D and calcium was very similar in patients with and without DR [58].

Millen et al. studied patients from the Atherosclerosis Risk in Communities study (ARIC), who had assessments of serum levels of vitamin D, dietary vitamin D intake, and retinal photography grading for DR severity. They found that, after 3 years, serum concentrations of 25(OH)D ≥ 75 nmol/L were associated with lower odds of any DR, while no statistically significant association was found between vitamin D intake from foods or supplements and DR [52].

## 4. Discussion

Since oxidative stress has been found to play roles in the pathogenesis of DR, vitamins, which generally have anti-oxidative effects, should theoretically be used as a treatment modality for DR. However, based on the published data in the literature, the roles of vitamins in DR have not yet been well-established, as in AMD, another retinal disease which is also a leading cause of blindness. The U.S. Food and Drug Administration (U.S. FDA) classifies vitamins as dietary supplements, although when the intent to use vitamins is for prevention or treatment of diseases, they should be considered drugs. Vitamins may be necessary for people who may not have enough dietary intake of essential nutrients, but they are generally used to improve. or maintain, overall health. The U.S. FDA does not approve dietary supplements, including vitamins. but regulates their marketing, inspects their manufacturing facilities, and monitors their safety [74].

Research on the roles of dietary vitamin supplements in other diseases, such as cardiovascular disease [7,75], cancer [7,76], and neurodegenerative disease have been published [9]. All these diseases, and DR, are multifactorial; hence, measuring certain parameters, such as the amount of dietary vitamin intake, in these diseases is difficult. The baseline dietary intake for individuals with the same severity level of these diseases may still be different. In addition, the dietary intake of healthy individuals in different countries could be different. Although the FFQ is used as a standard tool to estimate food and nutrient intake in individuals, adjustment according to countries is a common practice, since the FFQ also relies on the food composition database (FCD) of each country [70]. Being a subjective test, dependent on questionnaire answers, the FFQ is prone to bias, which could explain the inconsistent findings of the effect of dietary vitamins in these diseases [9,75,76,77].

A FCD is essential to measure the amounts of vitamin intake. However, there might be some variations between each national FCD. These variations could be due to many reasons, such as environmental influence on the ingredients, different breeds, processing of the food, different recipes, or cooking methods. Therefore, FCDs from different countries might result in slightly different amounts of nutrients or vitamins for the same dish. Standardized methods for cooked food in FCDs is important for comparison between each FCD [70]. Additionally, it has been proposed by Charrondiere et al. that, for compilation of reliable FCDs, the following pillars must be considered: international standard guidelines and tools for creating FCD must be developed and used, national FCDs should be regularly updated, and human resources must be trained in relation to food compositions [78].

Measuring serum levels of individual vitamins may be an objective method to find their associations with diseases, although the serum levels may not truly reflect the dietary intake of an individual vitamin. There are many measurement techniques and devices used to evaluate serum vitamin levels, for example, high-performance liquid chromatography (HPLC), enzyme-linked immunoassay (ELISA), radioimmunoassay, etc. The different methods or devices could cause variations in the results [79,80,81].

The larger number of studies on vitamin D and DR, compared with other individual vitamins, may not only be due to its potential as an inhibitor of retinal neovascularization but may also reflect the fact that vitamin D deficiency is considered one of the most common medical conditions worldwide, affecting approximately one billion people. Vitamin D has been studied more than other individual vitamins in general, not only in DR. There have been studies showing possible positive effects of vitamin D_3_ on retinal cells in animals with DR [36,37,38], which seemed to hint at what could be expected in humans, and, therefore, encouraged many studies on vitamin D in DR. Regarding DR, while a number of studies found lower serum levels of vitamin D among patients with DR, compared to those without DR [32,43,44,45,46,47,48,50], other studies found no association between vitamin D serum levels and DR status [28,42,57,58,59,61,62,63]. There were four meta-analyses on serum vitamin D levels and DR, and each concluded that there was an association between low serum vitamin D levels and increased risk of development of DR, but there was high heterogenicity in the studies [51,82,83,84]. There have not yet been meta-analyses of the association between serum levels of other individual vitamins and DR. Since the number of studies of each individual vitamin, such as A [17,18], B [25,26,27,28,29], C [32,33,34,35], and E [32,34,35,66], is small, and comparison between them for each vitamin is difficult, the associations between serum levels of these vitamins and DR remains inconclusive.

Of all studies on vitamins for DR, there were two studies, an RCT by Ho et al. and a pilot study by Smolek et al., which intended to use vitamins as drugs for treatment. In the RCT, the authors aimed to use potent anti-oxidative effects, which might also have potential in angiogenesis inhibition, for treatment of DR. This study used areas of retinal hemorrhage and DME from retinal photographs, in the format of standard Early Treatment Diabetic Retinopathy Study 7-field, but not stereoscopic [85], as the outcome measures. Assessment of retinal hemorrhage, one of many retinal lesions in DR, alone might not be a good surrogate for assessment of DR progression. In addition, assessment of DME from retinal photographs may not be as reliable as from optical coherence tomography (OCT) [67]. In the pilot study, the authors aimed to study vitamin B, which was found to lower the risk of cardiovascular diseases and nephropathy, in the treatment of DR. The retinal sensitivity outcomes, measured by microperimetry and retinal thickness by OCT in this study, may be more acceptable than the outcomes in the RCT by Ho et al. However, the very small sample size was a main limitation of this study [68]. Until now, the results from the study, that vitamin B can increase retinal sensitivity or decrease retinal thickness, have not yet been found in other studies.

Although two large prospective cohort studies with long-term follow-ups from Japan concluded that high dietary intake of vitamin B6, and fruit rich in vitamins C and E were associated with a decreased risk of DR development [71,72], further RCTs are necessary to clarify and validate the benefit of high intake of these vitamins.

Strategies for prevention in medicine include primary, secondary, and tertiary strategies to prevent disease from occurring, from worsening, and from causing permanent morbidity or mortality, respectively. For DR, primary prevention is for patients with diabetes without retinopathy, secondary prevention is for patients with DR without sight-threatening DR, and tertiary prevention is for patients with sight-threatening DR [86]. In this regard, few studies of vitamins for DR showed potential for primary prevention, fewer studies for secondary prevention, and none for tertiary prevention.

## 5. Conclusions

The purpose of vitamin intake with regard to DR is to decrease the risk of development, or progression, of DR. The intake can be in the form of food or vitamin tablets. Vitamin from food can be measured in the amounts of milligrams per day, using the standard FFQ protocols, whereas serum levels of each vitamin can be measured directly. Although this review did not find strong evidence of the relationship between vitamin intake and the development, or progression, of DR, large prospective cohort studies have provided evidence that the long-term intake of vitamin-rich fruit or food with high vitamin B6 may lower the risk of development of DR. Since dietary control is part of the management of DM, this information may be given to patients as part of current evidence on dietary recommendations for patients with diabetes. There is potential association between vitamin D deficiency, measured as serum level, and the development, or progression, of DR. However, these results have not been supported by studies on dietary intake of vitamin D and DR. Future research on vitamins and DR may focus on RCT with a proper study design, adequate sample size, and a long enough follow-up period. These studies may provide stronger evidence to support observational studies on the intake of vitamin-rich fruit and foods with high vitamin B and vitamin D contents.

## Figures and Tables

**Figure 1 jcm-11-06490-f001:**
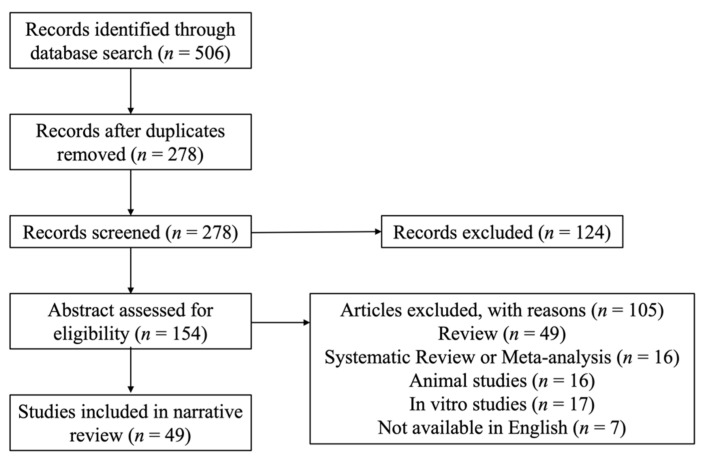
Selection of included studies.

**Table 1 jcm-11-06490-t001:** Studies on association between circulating vitamin levels and diabetic retinopathy.

Reference	N (Subjects)	Measurement	Primary Outcome
**Vitamin A**			
Zhang et al.	126 (43 T2DM without DR, 43 T2DM with DR, 40 healthy controls)	- HPLC technique for serum vitamin A level measurement	- Serum vitamin A levels were lower in diabetic participants without retinopathy than in the DR group (*p* < 0.001). - There were no differences in serum vitamin A levels between participants with DR and the healthy control group.
Rostamkhani et al.	60 (20 no DR, 20 NPDR, 20 PDR)	- HPLC technique for serum vitamin A level measurement	- Patients with DR had significantly lower serum vitamin A levels than control (*p* = 0.01). - Higher serum vitamin A level reduced the risk of the development of DR by 31.1% (*p* = 0.007).
**Vitamin B**			
Satyanarayana et al.	394 (100 no DR, 194 DR, 100 healthy controls)	- HPLC method for plasma vitamin B1, B2, and B6 measurement - Radioimmunoassay for plasma vitamin B9 and B12 measurement	- No significant difference in plasma vitamin B1 levels between diabetic patients with and without retinopathy. - Plasma vitamin B6 and B12 levels were significantly lower in the diabetes group than in the control group (*p* < 0.05).
Srivastav et al.	80 (20 no DR, 20 NPDR with DME, 20 PDR with DME, 20 healthy controls)	- Elecsys Folate III was used for serum vitamin B9 measurement - Elecsys Vitamin B12 was used for serum vitamin B12 measurement	- Serum levels of both vitamin B9 and B12 in any groups were not different from healthy controls.
Cinici et al.	100 (20 no DR, 20 mild-moderate NPDR, 20 severe NPDR, 20 PDR, 20 healthy controls)	- HPLC method for thiamine pyrophosphate measurement	- Patients with any DR stages had lower TPP levels than healthy controls (*p* < 0.05).
Tomic et al.	94 (69 no DR, 25 NPDR)	- Automated immunoturbidimetric procedure on a dedicated analyzer	- Serum vitamin B12 levels in patients with NPDR were lower than in the no DR group (*p* = 0.028). - Serum vitamin B9 levels were not different between no DR and NPDR groups.
Malaguarnera et al.	311 (96 no DR, 70 NPDR, 65 PDR, 80 healthy controls)	- Quantaphase II folate radioassay kit	- Serum vitamin B9 levels in patients with PDR were significantly lower than no DR, NPDR and healthy controls (*p* < 0.01, <0.05 and <0.01, respectively).
**Vitamin C**			
Kumari et al.	112 (30 no DR, 42 DR, 40 healthy controls)	- Conversion to dehydro ascorbic acid and measuring at 520 nm in spectrophotometer	- Serum vitamin C levels in the DR group were significantly lower than those without DR and healthy controls.
Kundu et al.	150 (50 DM without DR, 50 DR, 50 healthy controls)	- Measurement method was not mentioned in the article.	- Serum vitamin C levels were significantly lower in the DR group than the DM without DR group and healthy controls (*p* < 0.001).
Lam et al.	420 (46 no DR, 161 background DR, 207 NPDR, 6 PDR)	- FRAP assay for determining plasma vitamin C concentration	- Plasma vitamin C concentration was not different between any of the groups.
Longo-Mbenza et al.	200 (150 T2DM, 5 non-DM with retinopathy, 45 healthy control)	- HPLC with multiwavelength was used to measure serum vitamin C level	- Serum vitamin C levels were significantly different between non-diabetics without DR (controls), T2DM patients without DR and T2DM patients with DR (<0.0001). DM patients with DR had the lowest level.
**Vitamin D**			
Ahmed et al.	750 (460 T2DM, 290 healthy controls)	- Isotope dilution liquid chromatography tandem mass spectrometry was used to measured serum vitamin D levels.	- The lower serum 1,25(OH)_2_D_3_ and 25(OH)D_3_ levels were associated with DR (*p* = 0.006, *p* = 0.03, respectively). - There was no association between serum 24,25(OH)_2_D or 24,25(OH)_2_D_3_ with DR.
Tomic et al.	94 (69 no DR, 25 NPDR)	- Automated immunoturbidimetric procedure on a dedicated analyzer	- Serum vitamin D levels were not different between no DR and NPDR groups.
Afarid et al.	60 (30 no DR, 30 DR)	- Solid phase ELISA was used to measure serum 25-OH vitamin D level.	- The mean serum vitamin D level was lower in the DR group than the non-DR group (*p* = 0.012). - No difference in serum vitamin D level among the DR subgroup.
Lopes et al.	182 T1DM (79 no DR, 103 DR)	- Serum total 25(OH)D levels were retrospectively obtained from clinical records.	- No difference in serum total 25(OH)D levels between T1DM patients with and without retinopathy.
Alcubierre et al.	283 T2DM (144 no DR, 139 DR)	- Chemiluminescent microparticle immunoassay for serum vitamin D measurement	- No significant difference between serum vitamin D levels for patients with DR and control.
Yuan et al.	889 (616 no DR, 273 DR)	- Electrochemiluminescence method was used for serum vitamin D level.	- Vitamin D deficiency was associated with higher risk of DR (OD 1.84, 95%CI 1.18–2.86).
Nadri et al.	72 (24 no DR, 24 NPDR, 24 PDR)	- Chemiluminescence delayed, one-step assay was used for serum 25(OH)D concentration measurement.	- Serum vitamin D levels were significantly lower in the PDR group than those in NPDR, no DR and healthy controls (*p* < 0.001)
Longo-Mbenza et al.	200 (150 T2DM, 5 non-DM with retinopathy, 45 healthy control)	- HPLC with multiwavelength was used to measure serum vitamin D level	- Serum vitamin D levels were significantly different between non-diabetics without DR (controls), T2DM patients without DR and T2DM patients with DR (<0.0001). DM patients with DR had the lowest level.
Ashinne et al.	3054 (1647 no DR, 1174 NPDR, 232 PDR)	- Electrochemiluminescence immunoassay	- Serum 25(OH)D was lower in the people with DR than in those without DR (11.9 ± 2.2 vs. 13.7 ± 2.1 ng/mL, *p* < 0.001).
Long et al.	842 T2DM (541 no DR, 195 mild NPDR, 106 severe NPDR&PDR)	- Radioimmunoassay - Liquid chromatography-tandem mass spectrometry	- VDD was associated with increased odds of DR development (OR 2.226 95%CI 1.359–3.648, *p* = 0.002).
Zoppini et al.	715 T2DM (490 no DR, 168 NPDR, 57 DR)	- Chemiluminescence immunoassay	- The group with vitamin D insufficiency had higher prevalence of DR than the other group with sufficient vitamin D levels (*p* = 0.018).
Alam et al.	657 T2DM (257 no DR, 243 background DR, 135 pre PDR, 22 PDR)	- Chemiluminescence immunoassay	- No difference in serum vitamin D level between non-DR and DR groups.
Shimo et al.	75 T1DM	- Radioimmunoassay	- VDD was associated with DR (OR 3.45, 95%CI 1.11–10.6, *p* < 0.03).
Millen et al.	1339 T2DM: 280 PDR (21%)	- Liquid chromatography in tandem with high-sensitivity mass spectrometry	- Participants with adequate serum vitamin D level had lower odds of DR than those with insufficient vitamin D (*p* trend < 0.001).
Usluogullari et al.	311 (73 no DR, 238 DR)	- HPLC method	- No difference in serum vitamin D level between non-DR and DR groups.
He et al.	1520 T2DM (625 no DR, 562 non-STDR, 333 STDR)	- Electrochemiluminescence assay	- Patients with DR had lower serum vitamin D levels than those without DR (*p* < 0.05).
Jee et al.	2113 T2DM (1738 no DR, 375 DR)	- Radioimmunoassay kit and a gamma counter	- No association between serum vitamin D level and DR status.
Reddy et al.	263 (82 no DR, 82 DR, 99 healthy controls)	- HPLC method	- No difference in mean serum vitamin D levels between no DR and DR groups.
Bajaj et al.	158 T2DM	- The method of measurement was not disclosed.	- VDD was associated with DR (*p* = 0.010).
Bonakdaran et al.	235 (153 no DR, 64 NPDR, 18 PDR)	- Radioimmunoassay method	- Serum vitamin D levels were not different between non-DR and DR groups.
Payne et al.	221 (47 no DM, 51 no DM with ocular disease, 41 DM without DR, 40 NPDR, 42 PDR)	- Automated competitive chemiluminescence immunoassay	- Serum vitamin D levels were significantly lower in those with either NPDR or PDR than healthy control (*p* = 0.015, and 0.002, respectively).
Kaur et al.	517 T1DM	- Automated chemiluminescence analyzer	- Retinopathy was more common in participants with VDD (*p* = 0.02).
**Vitamin E**			
Kumari et al.	112 (30 no DR, 42 DR, 40 healthy controls)	- Baker and Frank method	- Serum vitamin E levels were significantly lower in the DR group than the DM without DR group, and healthy control.
Said et al.	128 (30 healthy controls, 30 DR without retinopathy, 34 NPDR, 34 PDR)	- Measurement method was not described in the article.	- Serum vitamin E levels were significantly decreased in patients with retinopathy than those without (*p* < 0.0001).
Longo-Mbenza et al.	200 (150 T2DM, 5 non-DM with retinopathy, 45 healthy control)	- HPLC with multiwavelength was used to measure serum vitamin E level	- Serum vitamin E levels were significantly different between non-diabetics without DR (controls), T2DM patients without DR and T2DM patients with DR (<0.0001). DM patients with DR had the lowest level.
Lam et al.	420 (46 no DR, 161 background DR, 207 NPDR, 6 PDR)	- FRAP assay for determining plasma vitamin C concentration	- Plasma vitamin E levels were not different between those with no DR, NPDR, and PDR groups.

DR = Diabetic retinopathy, ELISA = Enzyme-linked immunosorbent assay, HPLC = High Performance Liquid Chromatography, MD = Mean difference, NPDR = Non-proliferative diabetic retinopathy, PDR = Proliferative diabetic retinopathy, STDR = sight-threatening diabetic retinopathy, T1DM = Type 1 diabetes mellitus, T2DM = Type 2 diabetes mellitus, VDD = Vitamin D deficiency.

**Table 2 jcm-11-06490-t002:** Studies on associations between dietary intakes of vitamins and diabetic retinopathy.

Reference	N (Subjects)	Study Design	Measurement Methods	Primary Outcomes
**Vitamin A**				
Zhang et al.	126 (43 no DR, 43 DR, 40 healthy controls)	- Cross-sectional study	- FFQ based on food groups was used. - Dietary intake of nutrients was calculated using the Nutrition System of Traditional Chinese Medicine Combining with Western Medicine, version 11.0	- Dietary vitamin A levels were significantly higher in no DR group than DR and healthy control groups (*p* < 0.05). - Dietary intake of 100 µg/day of vitamin A was associated with 17% lower risk of DR (OR 0.83, 95%CI 0.70–0.98).
**Vitamin B**				
Horikawa et al.	978 T2DM no DR from Japanese Diabetic Complication Study	- Prospective observational study as a part of the Japan Diabetes Complication Study (JDCS)	- FFQ based on food groups was used. - Dietary intake of nutrients was calculated using Standard Tables of Food Composition in Japan	- Patients with T2DM who consumed 2.0 mg/day of vitamin B6 had 50% lower risk of DR than those who consumed 0.9 mg/day (*p* = 0.010).
Smolek et al.	10 T2DM with bilateral mild-moderate NPDR	- Prospective, open-label, single-arm pilot study	- Participants were given vitamin tablets containing 3 mg L-methylfolate calcium, 35 mg pyridoxal-5-phosphate, and 2 mg methylcobalamin twice daily.	- Nonlinear improvement in mean threshold retinal sensitivity (*p* < 0.001) - Significant reduction in mean central retinal thickness between one and six months (*p* = 0.012)
**Vitamin C**				
Tanaka et al.	978 T2DM no DR from Japanese Diabetic Complication Study	- Prospective observational study as a part of the Japan Diabetes Complication Study (JDCS)	- FFQ based on food groups was used. - Dietary intake of nutrients was calculated using Standard Tables of Food Composition in Japan	- Intake of fruits rich in vitamin C showed a protective effect against DR with a HR (95% CI) of 0.61 (0.39–0.96).
Sasaki et al.	379 T2DM (150 no DR, 142 NPDR, 87 PDR)	- Cross-sectional study	- A semi-quantitative FFQ was used. - Dietary intake of nutrients was calculated using the Australian Tables of Food Composition.	- No association between vitamin C intake and DR in well-controlled DM patients.
**Vitamin D**				
Alcubierre et al.	283 (144 no DR, 139 DR)	- Observational case-control study	- FFQ based on food groups was used. - Dietary intake of nutrients was calculated using food composition tables of the US Department of Agriculture.	- No statistically significant association was found between vitamin intake and DR.
Millen et al.	1339 (207 mild NPDR, 44 moderate to severe NPDR, 29 PDR, 3 DME)	- Prospective observational study as a part of Atherosclerosis Risk in Communities (ARIC) study	- Willett 66-item semi-quantitative FFQ was used.	- No associations were found between vitamin D intake and DR.
**Vitamin E**				
Tanaka et al.	978 T2DM no DR from Japanese Diabetic Complication Study	- Prospective observational study as a part of the Japan Diabetes Complication Study (JDCS)	- FFQ based on food groups was used. - Dietary intake of nutrients was calculated using Standard Tables of Food Composition in Japan	- No significant association between dietary vitamin E intake and DR.
Sasaki et al.	379 T2DM (150 no DR, 142 NPDR, 87 PDR)	- Cross-sectional study	- A semi-quantitative FFQ was used. - Dietary intake of nutrients was calculated using the Australian Tables of Food Composition.	- No association between dietary vitamin E intake and DR.
Ho et al.	55 NPDR	- Multicenter, prospective, double-blinded, placebo-controlled, RCT	- RCT comparing between tocotrienol-rich vitamin E supplement and placebo supplementation for 12 months.	- No significant difference in percentage changes in area of retinal hemorrhage or DME at 2, 6, and 12 months of treatment.

DM = Diabetes mellitus, DME = Diabetic macular edema, DR = Diabetic retinopathy, FFQ = Food frequency questionnaire, HR = Hazard ratio, OR = Odd ratio, RCT = Randomized controlled trial, SFFQ= Semiquantitative food frequency questionnaire, WMD = Weighted mean difference.

## Data Availability

Not applicable.

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
