# Peer review of "The Roles of Vitamins in Diabetic Retinopathy: A Narrative Review"

_jcm, 2022, doi:10.3390/jcm11216490_

Round 1

Reviewer 1 Report

This review is well organized and it seizes on a subject which is very important and really interesting.

I just have a couple of comments:

- did the authors considered the role and involvement of CYP24A1 and CYP27B1 (as regard as vitamin D3)?

- please, fix "1,25(OH)2D3" with "1,25(OH)2D3"

-please, try to simplify the tables, in order to be less confusing

-in the discussion section, at line 297, the authors could potentially discuss that the effects of vitamin D3 have been demonstrated in in vitro and in vivo models of diabetic retinopathy (i.e., PMID: 29682582; PMID: 36091806; PMID: 31236602)

Author Response

Response to Reviewer 1 Comments

This review is well organized and it seizes on a subject which is very important and really interesting.

I just have a couple of comments:

Point 1: Did the authors considered the role and involvement of CYP24A1 and CYP27B1 (as regard as vitamin D3)?
Response 1: We did not include CYP24A1 and CYP27B1 in this review. There was only one study on CYP24A1 and diabetic retinopathy and no studies on CYP27B1 and diabetic retinopathy were found. The published study on CYP24A1 was conducted in animal model.

Point 2: Please, fix "1,25(OH)2D3" with "1,25(OH)2D3"
Response 2: We have made the correction accordingly. (Line 166-168)

Point 3: Please, try to simplify the tables, in order to be less confusing
Response 3: We have edited both Table 1 and Table 2.

Point 4: In the discussion section, at line 297, the authors could potentially discuss that the effects of vitamin D3 have been demonstrated in in vitro and in vivo models of diabetic retinopathy (i.e., PMID: 29682582; PMID: 36091806; PMID: 31236602).
Response 4: We have included more details on animal and in vitro studies in the results section. (Line 149-152) We have also mentioned this matter in the Discussion section. (Line 320-323)

Reviewer 2 Report

Dear Authors, I have read your manuscript with interest.

The current manuscript titled: "The Roles of Vitamins in Diabetic Retinopathy: A Narrative Review" represents an important analysis of evolving field of Ophthalmology. There is currently a special attention on the diabetic retinopathy.

The title reflects the manuscript content and helps the reader navigate the article essence.

In my opinion, these are the adjustments which should be made to increase the value of your manuscript:

1.       In Introduction section, please describe information about diabetes mellitus first, then diabetic retinopathy and vitamins. Also, add more detailed information about diabetic retinopathy, including epidemiology and physiopathological mechanisms.

2.       In the Methodology section, indicate whether you searched for specific vitamins or only, as indicated, for the keyword Vitamins. If yes, add a search for all the studied vitamins separately. Also, add a graphical selection of articles.

3.       In Tables 1 and 2, change the location of the first column as the spelling of vitamins is not clear.

4.       The conclusions largely repeat the information presented in the main part of the article. There is no synthesis of information and practical conclusions that would help readers understand the purpose, utility and practical significance of this manuscript.

5.       The manuscript contains some punctuation errors and typos, please revise the text.

Author Response

Response to Reviewer 2 Comments

Dear Authors, I have read your manuscript with interest.

The current manuscript titled: "The Roles of Vitamins in Diabetic Retinopathy: A Narrative Review" represents an important analysis of evolving field of Ophthalmology. There is currently a special attention on the diabetic retinopathy.

The title reflects the manuscript content and helps the reader navigate the article essence.

In my opinion, these are the adjustments which should be made to increase the value of your manuscript:

Point 1: In Introduction section, please describe information about diabetes mellitus first, then diabetic retinopathy and vitamins. Also, add more detailed information about diabetic retinopathy, including epidemiology and physiopathological mechanisms.
Response 1: We have added more details on diabetes mellitus (Line 27-36), and also more information about diabetic retinopathy. (Line 42-52)

Point 2: In the Methodology section, indicate whether you searched for specific vitamins or only, as indicated, for the keyword Vitamins. If yes, add a search for all the studied vitamins separately. Also, add a graphical selection of articles.
Response 2: We did not search for the specific vitamins included since they are subsets of the query term used in this study. We have added a graphical selection of articles, and made some changes in the Methodology section to be more accurate. (Figure 1)

Point 3: Tables 1 and 2, change the location of the first column as the spelling of vitamins is not clear.
Response 3: We have edited both Table 1 and Table 2.

Point 4: The conclusions largely repeat the information presented in the main part of the article. There is no synthesis of information and practical conclusions that would help readers understand the purpose, utility and practical significance of this manuscript.
Response 4: We have re-written the Conclusion section. (Line 362-377)

Point 5: The manuscript contains some punctuation errors and typos, please revise the text.
Response 5: We have checked through the manuscript and made the correction accordingly.

Round 2

Reviewer 2 Report

I agree with the changes made, which significantly improve the quality of the manuscript. I recommend this article for publication. Good luck!